# Phylogenetic Groups, Pathotypes and Antimicrobial Resistance of *Escherichia coli* Isolated from Western Lowland Gorilla Faeces (*Gorilla gorilla gorilla*) of Moukalaba-Doudou National Park (MDNP)

**DOI:** 10.3390/pathogens11101082

**Published:** 2022-09-23

**Authors:** Leresche Even Doneilly Oyaba Yinda, Richard Onanga, Pierre Philippe Mbehang Nguema, Etienne François Akomo-Okoue, Gontran Nsi Akoue, Neil Michel Longo Pendy, Desire Otsaghe Ekore, Roméo Wenceslas Lendamba, Arsène Mabika-Mabika, Jean Constant Obague Mbeang, Natacha Poungou, Jacques François Mavoungou, Sylvain Godreuil

**Affiliations:** 1Laboratory of Bacteriology, Interdisciplinary Medical Research Center of Franceville, Franceville P.O. Box 769, Gabon; 2Microbiology Laboratory, Research Institute for Tropical Ecology, Libreville P.O. Box 13354, Gabon; 3École Normale Supérieure (ENS), Libreville P.O. Box 17009, Gabon; 4Laboratory of Vector Ecology, Interdisciplinary Medical Research Center of Franceville, Franceville P.O. Box 769, Gabon; 5Laboratory of Biology, University of Science and Technology of Masuku, Franceville P.O. Box 913, Gabon; 6Laboratoire de Bactériologie, CHU de Montpellier, UMR MIVEGEC (IRD, CNRS, Université de Montpellier), 34295 Montpellier, France

**Keywords:** *E. coli*, antimicrobial resistance, phylogenetic group, pathotype, gorilla, MDNP, Gabon

## Abstract

(1) Background: Terrestrial mammals in protected areas have been identified as a potential source of antimicrobial-resistant bacteria. Studies on antimicrobial resistance in gorillas have already been conducted. Thus, this study aimed to describe the phylogroups, pathotypes and prevalence of antimicrobial resistance of *Escherichia coli* isolated from western lowland gorilla’s faeces living in MDNP. (2) Materials and Methods: Ninety-six faecal samples were collected from western lowland gorillas (*Gorilla gorilla gorilla*) during daily monitoring in the MDNP. Sixty-four *E. coli* isolates were obtained and screened for phylogenetic and pathotype group genes by polymerase chain reaction (PCR) after DNA extraction. In addition, antimicrobial susceptibility was determined by the disk diffusion method on Mueller Hinton agar. (3) Results: Sixty-four (64%) isolates of *E. coli* were obtained from samples. A high level of resistance to the beta-lactam family, a moderate rate for fluoroquinolone and a low rate for aminoglycoside was obtained. All *E. coli* isolates were positive in phylogroup PCR with a predominance of A (69% ± 11.36%), followed by B2 (20% ± 19.89%) and B1 (10% ± 8.90%) and low prevalence for D (1% ± 3.04%). In addition, twenty *E. coli* isolates (31%) were positive for pathotype PCR, such as EPEC (85% ± 10.82%) and EPEC/EHEC (15% ± 5.18%) that were obtained in this study. The majority of these MDR *E. coli* (DECs) belonged to phylogenetic group A, followed by MDR *E. coli* (DECs) belonging to group B2. (4) Conclusion: This study is the first description of MDR *E. coli* (DECs) assigned to phylogroup A in western lowland gorillas from the MDNP in Gabon. Thus, wild gorillas in MDNP could be considered as asymptomatic carriers of potential pathogenic MDR *E. coli* (DECs) that may present a potential risk to human health.

## 1. Introduction

Many zoonoses are associated with ecological imbalances linked to deforestation, human population expansion, changes in agricultural practices and encroachment on wildlife habitats [1]. The intensification of these human activities within previously isolated wildlife habitats is a key factor in the emergence of infectious diseases [2,3]. Certainly, the spread of zoonotic diseases in human populations is a phenomenon on which particular attention has been paid [4,5,6]. On the other hand, the ever-increasing anthropization of wildlife habitats also favours the emergence of diseases in wildlife [2,7].

These zoonoses, caused by pathogenic microorganisms such as enterobacteria (such pathogenic *E. coli*), have been a human public health problem for decades [8,9]. Indeed, many studies have shown that wildlife [3,5,10,11,12,13], including wild primates, could transmit pathogens to humans [14,15,16,17]. Pathogenic *E. coli* strains belong to phylogenetic groups B2 and D [18,19,20], while commensal or wild-type strains, associated with phylogenetic groups B1 and A, are more frequently isolated in environment [21,22]. Pathogenic *E. coli* (DECs) are classified as enteropathogenic (EPEC), enterohemorrhagic (EHEC), enterotoxigenic (ETEC), entero-invasive (EIEC), enteroaggregative (EAEC) and diffuse adhesion (DAEC) *E. coli* [23].

Indeed, while habitat loss and poaching are recognized threats to the survival of great apes [24,25], infectious diseases are emerging as the real challenge facing the conservation of wildlife biodiversity. For example, the listing of the western lowland gorilla (*Gorilla gorilla gorilla*) in the IUCN Red List of Threatened Species (Version 2015.4. www.iucnredlist.org, accessed on 31 March 2022) is a consequence of the great apes massacre caused by the Zaire strain of the Ebola virus; thousands of common chimpanzees (*Pan troglodytes troglodytes*) and lowland gorillas in Central Africa were indeed decimated during this epidemic [26,27,28].

In addition, multidrug-resistant (MDR) pathogens, mainly enterobacteria, have been associated with increased mortality rates in humans and animals, including non-human primates (NHPs) [29]. Indeed, the close phylogenetic relationship between humans and other human species such as non-human primates, combined with the expansion of the human–animal interface, facilitates the transmission of pathogens between species, leading to morbidity and mortality in great ape populations worldwide [14,15,30,31,32,33,34,35,36].

Many substantiated cases of transmission of human pathogens to non-human primates [37], including chimpanzees [14] and mountain gorillas [15], have resulted from interactions (hunting, ecotourism, research activities, etc.) between humans and non-human primates.

Indeed, exchange of *E. coli* between humans, domestic animals and apes has been reported in densely populated areas of western Uganda. In Uganda, habituated groups of wild apes are visited daily by researchers and tourists, such as chimpanzees in Kibale [38] and mountain gorillas in Bwindi Impenetrable National Park [39].

In the forest of Moukalaba-Doudou National Park (MDNP), Gabon, ecotourism is being introduced due to the human habituation of groups of western lowland gorillas (*Gorilla gorilla gorilla*) [40,41]. The protection of human and animal populations, especially great apes, from mutual transmission of epidemics is therefore important [42].

*Enterobacteriaceae*, particularly *Escherichia coli*, *Klebsiella* spp., *Enterobacter cloacae*, *Proteus mirabilis* and *Serratia* spp., carrying antimicrobial resistance or multidrug-resistant Enterobacteriaceae (MRAE), have already been isolated from wild mammals in protected areas [43,44] and unprotected areas of Gabon [45], with a lower prevalence of bacteria isolated from wildlife compared to humans. Among these EMRAs, EBLSEs have recently been isolated from fruit bats captured in highly anthropized urban areas [45]. The level of antimicrobial-resistant bacteria in wildlife, particularly in western lowland gorillas (*Gorilla gorilla gorilla*), could reveal the true level of the anthropological impact on wildlife in the PNMD.

In Gabon, studies on antimicrobial resistance of enterobacteria isolated from gorillas in the Lopé National Park [46] and in the MDNP [43,44] have already been conducted. It has also been shown that in protected areas wild types of *E. coli* carrying natural (intrinsic) resistances predominate over clinical strains carrying acquired resistances [47]. However, there is a lack of data on pathogenic antibiotic resistant *E. coli* in the wild mammals in protected areas. These pathogenic strains of antimicrobial-resistant or MDR *E. coli* are potential zoonoses that could be spread to humans [48,49,50,51] and create a real public health problem.

This study aims to describe the phylogenetic groups and pathotypes associated with antimicrobial resistance of *E. coli* isolated from faecal samples of gorillas living in MDNP (south-western Gabon).

## 2. Results

### 2.1. E. coli Found in Gorilla Faecal Samples

Following culture on solid media, isolation and biochemical identification, 64 *E. coli* isolates were obtained from 96 faecal samples collected from gorillas in the MDNP forest.

### 2.2. Prevalence of E. coli Phylogroups

Among the 64 *E. coli* isolates, phylogroup A was predominant (69% ± 11.36%; *n* = 44/64), followed by phylogroups B2 (20% ± 19.89%; *n* = 13/64), B1 (10% ± 8.90%; *n* = 6/64) and D (1% ± 3.04%%; *n* = 1/64), respectively (Table 1).

### 2.3. Prevalence of E. coli Pathotypes

Characterization of *E. coli* pathogenicity genes showed that of the 64 isolates 20 (31% ± 11.36%) carried these genes. Among pathogenic isolates, there was a predominance of the EPEC pathotype (85% ± 10.82%; *n* = 17/20). Three isolates (15% ± 5.18%; *n* = 3/20) were positive for the EPEC/EHEC complex pathotype (Table 2).

### 2.4. Antibiotic Susceptibility Testing

From the 64 *E. coli* isolates, 62 (97% ± 4.26%) were resistant to at least one antimicrobial agent, and resistance was most frequently observed against the beta-lactam family, such as amoxicillin-clavulanic acid (60% ± 12.03%) followed by imipenem (54% ± 12.23%), ticarcillin (41% ± 12.03%), cefotetan (41% ± 12.03%), piperacillin/tazobactam (36% ± 11.76%), piperacillin (34% ± 11.64%), cefoxitin (33% ± 11.50%) and ertapenem (8% ± 6.57%). In the aminoglycoside family, isolates were resistant to gentamycin (6% ± 7.14%), to tobramycin (5% ± 1.24%), to netylmicin (5% ± 1.24%) and to amikacin (3% ± 5.18%). In the quinolones and fluoroquinolones family, isolates were resistant to nalidixic acid (19% ± 18.49%), to ciprofloxacin (10% ± 7.65%) and to levofloxacin (2% ± 4.26%). In the family of phosphonic acid, isolates were resistant to fosfomycin (11% ± 9.24%). In the family of phenicols, isolates were resistant to chloramphenicol (6% ± 7.14%). Isolates were also resistant to sulfamethoxazole (6% ± 7.14%), to nitrofurantonin (6% ± 7.14%), to tetracycline (6% ± 7.14%) and to levofloxacin (2% ± 4.26%); twelve of the 62 (19% ± 4.26%) were MDR (Figure 1).

## 3. Discussion

Studies have suggested that the habituation process of wild animals, including gorillas, and tourism activities in their home range are aggravating factors for increased disease transmission mechanisms between humans and wildlife [54,55]. Indeed, the presence of human observers stresses the animals, which increases their susceptibility to microbial diseases [54,55]. In addition, the increase in human–wildlife interactions, including gorillas, greatly facilitates the exchange of potential pathogens [56].

This study is the first identification of MDR *E. coli* (DECs) isolates assigned to different phylogenetic groups in a wild gorilla population in the PNMD in Gabon. Indeed, previous studies in Gabon have reported the prevalence of antimicrobial resistance in *Enterobacteriaceae*, including *E. coli*, isolated from the faeces of unprotected wild mammals from peri-urban areas [45,57], and the faeces of wild mammals from protected areas such as those in the PNMD [43,44] and the Lopé National Park [46]. In addition, studies conducted in the Lopé National Park have characterized *E. coli* isolates belonging to phylogenetic groups A, B1, B2 and D [46].

### 3.1. Isolation of E. coli in Wild Gorillas from MDNP

The presence of antimicrobial resistance in Gabon’s national parks has already been described. This study aimed to describe the phylogenetic groups, pathotypes and prevalence of antimicrobial resistance in *E. coli* isolated from gorillas living in MDNP in Gabon. In our study, the prevalence of *E. coli* was similar to that reported by Mbehang et al. [45] and Benavides et al. [46] in the gorilla population living in national parks in Gabon. *E. coli* is one of the enterobacteria frequently described in wildlife [58,59,60].

### 3.2. Prevalence of Phylogenetic Groups

The phylogenetic clustering method used in this study detected and classified *E. coli* isolated from gorilla faeces into four groups (A, B1, B2 and D) [52,53]. Groups B2 and D are associated with extraintestinal pathogenic isolates causing diarrheal diseases in humans and animals [18,19,20], while groups A and B1 are associated with commensal isolates [52,53]. However, group B2 and D isolates are less frequently isolated from the environment [59] or fish, frogs and reptiles compared to group A or B1 isolates [22].

Our results revealed the predominance of phylogroups A (44/69%) and B2 (13/21%) and are similar to those obtained by Benavides et al. [46]. Although the number of commensal *E. coli* isolates is greater than the number of pathotype-carrying *E. coli* isolates in our results, the finding of the latter would suggest that these wild gorillas are asymptomatic carriers of potentially pathogenic *E. coli* [61]. The relatively high prevalence of group B2 isolates has previously been associated with herbivorous and omnivorous animals [23] and would suggest a potential transmission of *E. coli* strains from humans to these gorillas.

The phylogenetic grouping results obtained in the Taï National Park (Ivory Coast), namely group A (68%), followed by group B1 (18%), group D (16%) and group B2 (0.4%) [62], are very similar to those obtained by us. The predominance of phylogroup A was, as in the present study, in agreement with the results obtained by Carlos et al. [63], who concluded that group A is more prevalent in omnivorous hosts, while group B1 is typical of herbivorous hosts. Studies conducted at Minnesota State Zoo showed that phylogenetic groups A, B1, B2 and D were isolated from a captive population of NHPs being tested. In this study, group B1 was the most predominant [64]. These results are similar to those observed in our study on wild gorillas.

### 3.3. Prevalence of Pathotypes

Detection of *E. coli* pathotypes revealed that 20 out of 64 (31% ± 11.36%) isolates carried *E. coli* pathotypes represented mainly by the EPEC group (85% ± 10.82%) followed by the EPEC/EHEC complex (5% ± 5.18%). Similar results were reported by Kolappaswamy et al. [62]. These DECs are most often implicated in the development of diarrhoea in humans and are thought to contribute to the weakening of the host immune system [64,65,66,67,68]. The results of our study suggest that these gorillas are asymptomatic carriers of potentially pathogenic *E. coli* isolates.

The prevalence of EPEC pathotypes observed in our study is higher than that obtained in wild semi-captive monkeys in India [69]. In addition, EPEC was the most common pathotype isolated from captive and semi-captive populations of non-human primates [62,63,64,65,66,67,68,69,70,71,72]. Similarly, the results observed in our study corroborate those obtained by a team of researchers from the State of Minnesota in the United States for the majority of the primates studied living in a zoo, concerning EHEC and EPEC pathotypes [73]. Indeed, the EPEC pathotype is a major public health problem in developing countries, as it is a significant cause of childhood diarrhoea, sometimes fatal [74,75]. However, this pathotype has different effects on non-human primates, i.e., some react as asymptomatic carriers and others are susceptible and suffer from diarrhoea [70,71,76,77].

The presence of these DECs in the faeces of western lowland gorillas in MDNP suggests potential human transmission through defecation and waste of all kinds produced by the human populations surrounding this protected area, and that these wild animals are important reservoirs of *E. coli* isolates potentially pathogenic to humans.

### 3.4. Antibiotic Susceptibility

In our study, the antimicrobial resistance prevalence of *E. coli* isolates to beta-lactam antimicrobials, such as amoxicillin-clavulanic acid, imipenem, cefotetan, piperacillin and piperacillin-tazobactam, were observed. These results could be due to the contamination of food eaten by the gorillas. Indeed, wild animals, including gorillas in MDNP, can acquire antimicrobial resistant bacteria and AMR genes through improperly disposed antimicrobial chemicals and through environments contaminated with animal faeces and remains [78,79,80,81]. This result could also be explained to a lesser extent by other mechanisms for resistance acquisition, such as horizontal gene exchange among bacteria or naturally acquired resistance [46], or even through the adaptation and evolution of these pathogenic enterobacterial isolates in their environment [82] or the consumption of various foods [46,83]. The results obtained in a very recent study tend to correlate with our observations on the prevalence of antimicrobial resistance in wildlife. According to the authors, antimicrobial-resistant bacteria can contaminate soil and water systems and spread to wildlife [84]. Similar results were also observed in a study conducted in the Apuan Alps Regional Park (Tuscany, Italy) [85].

However, antimicrobial resistance was obtained for aminoglycosides, tetracycline, nitrofurantoin, sulfamethoxazole-trimethoprim and phenicol in our study. Similar results have been observed in studies of terrestrial mammals in protected [45] and unprotected [45,57] areas of Gabon. *E. coli* is naturally sensitive to these antimicrobials [16]. Therefore, the presence of these resistances in the digestive tract of gorillas could also be explained by the low human consummation of these antibiotics in that area.

The rates of resistance to ciprofloxacin obtained in our study are similar to those observed by Albrechtova et al. in Taï National Park [86]. Janatova et al. [87] obtained similar results in habituated gorillas (*Gorilla gorilla gorilla*) in the Dzanga-Sangha protected area in CAR, also describing the presence of these resistance genes [87]. Ciprofloxacin is a highly mobile hydrophilic compound, present in high concentrations in aquatic environments such as hospital effluents [88,89]. Despite its low stability with short half-lives, ciprofloxacin has regularly been detected in various aquatic environments such as groundwater and drinking water [88,90,91,92]. This could be partially explained by the wide use of this antimicrobial in the treatment of infections in human populations living around the PNMD [43,44,45]. Additionally, this resistance could be due to the intrinsic exchange of the resistome present in the digestive tract of these animals [82].

In addition, in *E. coli*, resistance to ciprofloxacin could result from various genetic variations [93,94,95]. Most strains unable to develop a favourable set of mutations in the quinolone resistance determining regions (QRDRs) must rely primarily on the overexpression of efflux pumps [96,97]. Only a few sequence types of *E. coli* would be able to develop multiple energetically favourable QRDR mutations [97].

The antimicrobial resistance observed in our study could also be related to antimicrobials used in human therapy and consumed locally. This hypothesis had already been proposed [43,44,46]. According to these authors, previous studies had already postulated that contact and subsequent transmission of antimicrobial-resistant bacterial isolates from highly resistant sources, such as humans or livestock, could explain the occurrence of antimicrobial resistance in wildlife [81,98,99,100,101].

The prevalence of antimicrobial resistance obtained in our study is higher than that observed in Bwindi Impenetrable National Park, where tourism and wildlife research activities are conducted. However, the results of this work suggest that habitat overlap between humans, livestock and mountain gorillas could influence the patterns of gastrointestinal bacterial exchange between the species [39]. The low rates of antimicrobial resistance in chimpanzees and monkeys in Taï National Park are thought to be attributed to the naturally evolving resistome, particularly because the monkeys have never been treated with antibiotics—evidence that there was no transmission of multidrug-resistant enteric bacteria, including *E. coli*, from humans to non-human primates in the park. Hygiene practices employed in the TNP would be effective in protecting Taï NHPs from acquiring multi-drug resistant enteric bacteria from humans [86]. In addition, the results obtained in the study of mountain gorillas in Kibale National Park, Uganda, for chloramphenicol, ciprofloxacin, tetracycline, trimethoprim-sulfamethoxazole and nalidixic acid were lower than our results, while the antimicrobial resistance results for chimpanzees from the same site were closer to ours [102].

The results obtained by Weiss et al. provided further evidence that humans and animals harbour bacteria with phenotypically similar (often identical) antimicrobial resistance profiles to commonly available and tested human antimicrobials [24,39]. However, the proportion of resistant isolates in animals was about three times lower than in humans, to ensure that isolates from domestic animals and wild primates had comparable proportions of resistant isolates. The most common multiple resistance patterns in *E. coli* isolates from humans were also the most common multiple resistance patterns in *E. coli* isolates from both livestock and wild primates [102].

However, the results of this work are not consistent with those obtained very recently in nature reserves, protected areas and national parks in the Gambia for a wide range of non-human primates regarding trimethoprim, sulfamethoxazole and tetracycline [103]. Increased contact between animal species greatly facilitates the potential exchange of pathogens [104]. Analyses by the research team also suggested an exchange of *E. coli* strains between humans and wild non-human primates [103].

The antimicrobial resistance rates obtained in our study were much higher than those observed in the work of Clayton et al. in a Minnesota State Zoo for gentamicin, nalidixic acid, tetracycline, sulfamethoxazole and trimethoprim in gorillas and other non-human primate species studied [73]. The results obtained in this study could be explained by the fact that the majority of the antimicrobials tested had only recently been used, except for ampicillin, which had historically been used in zoos to treat physical injuries, often suffered by humans [73]. However, the prevalence of resistance observed in another previous study of snub-nosed monkeys in zoos in China [105] is very close to the rates obtained in our study. Geographical location, zoological practices and the species of primates examined are many factors that could explain this phenomenon.

Antimicrobial resistance results for chloramphenicol, ciprofloxacin, gentamicin, tetracycline and trimethoprim in Kibale chimpanzees were lower than ours. However, the lack of appreciable resistance to ciprofloxacin, neomycin, gentamicin and ceftiofur in humans or chimpanzees already suggested that local antimicrobial use was responsible for the observed trends. It should be noted that no antimicrobials had ever been administered to the Kibale chimpanzees, but also that antimicrobials are freely available in Uganda and can be used indiscriminately. Thus, the presence of clinically resistant isolates in chimpanzees provided further evidence of the transmission of resistant bacteria or resistance-conferring genetics from humans to chimpanzees [38].

### 3.5. Association between Antimicrobial-Resistant DECs and Virulence in E. coli

Resistance profiles obtained in our study were diverse. Twelve *E. coli* isolates showed MDR with a resistance index between 0.22–0.59 in this study (Table 3). The results of this study reveal that the majority of MDR *E. coli* (DECs) isolates belong to group A and group B1, which group commensal isolates, followed by groups B2 and D, which group pathogenic isolates (Figure 2). Recent studies have reported that antimicrobial resistance and virulence factor carriage are linked, in *E. coli* populations, to a community setting [106,107]. Furthermore, these data suggested that while the co-occurrence of resistance and virulence in *E. coli* is partially due to antimicrobial selection pressure, the presence of virulence factors might explain it [106].

Ecological niche, life history and the propensity of causing disease are markers that characterize the different *E. coli* phylogroups. Thus, groups B2 and D are less frequently isolated from the environment [109]. Regarding human disease, *E. coli* isolates recovered from extra-intestinal body sites are more likely to belong to phylotypes B2 or D than to groups A or B1 [110,111]. Previous studies have revealed differences in the prevalence of phylogenetic groups within virulent extra-intestinal *E. coli* [112]. Although the two phylogenetic groups most commonly linked to extra-intestinal *E. coli* virulence are the aforementioned groups B2 and D, some reports have shown a high frequency of group A (46%) among *E. coli* causing UTIs in humans [112].

In addition, the existence of resistance diversity in terrestrial mammals, both genotypically and phenotypically, has been previously described in different protected areas in Africa, including Gabon [43,44,45,46], CAR [87], the Ivory Coast [86] and Uganda [38,39]. These results point to the potential for transmission of these MDR *E. coli* (DECs) from wildlife to humans living close to these areas or often cohabiting with these animals, or vice versa [38,39]. The results obtained in our study would support this hypothesis insofar as the majority of MDR DECs isolated from the faeces of MDNP gorillas belong to phylogenetic group A almost exclusively isolated from humans. This information highlights the existence of a health threat to these endangered animals living in protected areas.

## 4. Materials and Methods

Feld research authorization

The sampling of gorilla faeces in their living area in Moukalaba-Doudou National Park (Figure 3) was carried out in the case of a research study (field research authorization N° 003/20/DG/JBLD).

### 4.1. Study Area

Faecal samples were collected in the home range of habituated gorillas of MDNP (Figure 3) [113,114,115] from 27 February to 10 March 2020 using the non-invasive method as described in previous studies [44,45].

**Figure 3 pathogens-11-01082-f003:**
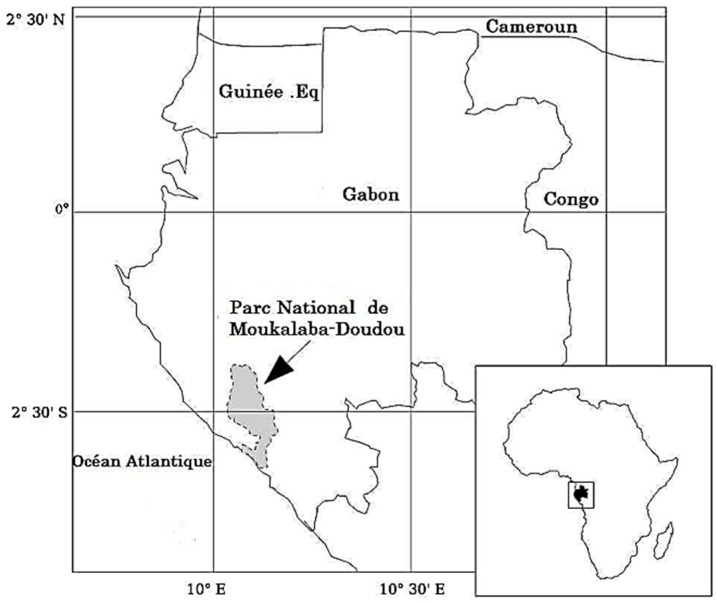
Location of the study area [116].

### 4.2. Sample Collection

Gorilla faeces were collected non-invasively by following gorillas in the forest.

Faeces left behind either following immediate defecation or three hours later were picked up based on the observation of their colour, temperature and consistency. To avoid bias from soil contamination, each faecal sample was collected inside the faecal material, avoiding surfaces exposed to the immediate environment and atmospheric oxygen, and the more reflective surfaces of fresh distal intestine (large intestine) samples using sterilized forceps. It is recommended that as many samples as possible be taken to ensure that a large number of individuals are sampled [43,44]. Each tube containing a faeces sample was stored in a bag in the shade and identified by a number which was transcribed on the tube and referenced in a small booklet next to which the identifiers of the said sample were marked (the diameter of the faeces, the collection area, the GPS point of the location, the name of the gorilla, etc.). Back at the research camp laboratory and after 3 to 4 h of sampling, a small portion of each sample was used to culture the bacteria and the remaining sample was stored/preserved in a solution of 70% physiological water (or phosphate-buffered saline (PBS)) and 30% glycerol. Samples preserved in such conditions could then be stored on the bench, in a bag or in a box, for more than two weeks at room temperature.

### 4.3. Culture, Isolation and Identification of Colonies

In the bacteriology laboratory of the Interdisciplinary Center for Medical Research of Franceville (CIRMF), the bacteria culture consisted of emulsifying some of the faecal specimens in sterile water and streaking a loopful of the resultant on MacConkey agar medium (MCA) (bioMérieux, France). After eighteen to twenty-four hours of incubation, several colonies (never more than two or three from one medium Petri dish), differentiated by structure and colour, were picked up and streaked in the same medium and incubated in the same conditions. A loop of liquid was removed from the cultures and streaked onto methylene eosin blue agar (EMB) plates (bioMérieux, France) for *E. coli* isolation. Plates were examined for the characteristic appearance of *E. coli* (metallic green sheen) and colonies with green metallic sheen were interpreted as *E. coli* isolates [117,118,119]. Purified green metallic colonies were subjected to biochemical identification using the Api 20 E galleries (bioMérieux, Marcy l’étoile, France) and the VITEK^®^ 2 Compact 15 system (bioMérieux, Marcy l’étoile, France). Isolates were confirmed by using phylogenetic group detection PCR as described by Clermont et al. [52,53].

### 4.4. Antimicrobial Susceptibility Testing of E. coli Isolates

Antimicrobial susceptibility testing was performed using the Kirby–Bauer disk-diffusion method [120]. The antimicrobial was chosen according to the recommendations of the Clinical Laboratory Standards Institute [121,122]. The antimicrobials tested were amoxicillin/clavulanic acid (AMC, 30 μg), ticarcillin (TIC, 75 µg), piperacillin (PIP, 100 μg), piperacillin/tazobactam (TZP, 100/10 µg), cefoxitin (FOX 30 μg), imipenem (IPM, 10 μg), ertapenem (ETP, 10 μg), amikacin (AMK, 30 μg), gentamycin (GEN, 10 μg), netilmicin (NET, 30 μg), tobramycin (TOB, 10 μg), chloramphenicol (CHL, 30 μg), tetracycline (TET, 30 μg), trimethoprim/sulfamethoxazole (SXT, 25 μg), nalidixic acid (NAL, 30 µg), ofloxacin (OFX, 5 μg), ciprofloxacin (CIP, 5 μg), levofloxacin (LVX, 5 µg), norfloxacin (NFX 10 µg), cefotetan (CTT 30 µg), fosfomycin (FOF, 200 g) and nitrofurantonin (NIT, 300 µg).

The breakpoints provided by the CLSI (2018) were used for the designation of isolates as resistant (R), intermediately susceptible (IS) or susceptible (S). For subsequent data analysis, the isolates with an I result were grouped with the isolates that gave an R result, and defined as resistant. Multidrug-resistant isolates were identified based on the definition of MDR as bacteria that are resistant to three or more classes of antimicrobial agents [84,123].

### 4.5. Molecular Identification of E. coli Phylogenetic and Pathotype Groups

#### DNA Extraction

DNA was extracted using the boiling method described by Mbehang Nguema et al. [45]. The characterization of the phylogenetic groups (A, B1, B2 and D) was carried out by amplification of the 3 genes chua, yjiaa and tspe4c2 in multiplex PCR as described by Clermont et al. [52,53] (Appendix A). The characterization of the pathotype groups was carried out under the conditions described by Sjöling et al. [124], using primers [125,126,127].

### 4.6. Data Analysis

The software Excel was used for data analysis, including calculations of the prevalence of *E.-coli*-carrying genes for detecting phylogenetic groups and pathotypes, the prevalence of antimicrobial resistance and confidence intervals. Adobe Illustrator software was used to design the histograms. The Multiple Antimicrobial Resistance Index was calculated following Krumperman [108]. All statistical analyses were performed using R 4.0.2. To visualize the data, a correspondence analysis was performed with the packages FactoMineR and Factoextra. In order to determine the relationship between the occurrence of EPEC virulence factors, antimicrobial resistance and phylogroups, a generalized linear binomial family model was generated with the stats package version 3.6.2 using the glm function. The best model illustrating this relationship was the one with ΔAIC = 0.

## 5. Conclusions

This study aimed to describe the phylogenetic groups and pathotypes associated with antimicrobial resistance of *E. coli* isolates from wild western lowland gorillas in the MDNP, a protected area in Gabon. The presence of these MDR *E. coli* (DECs), observed in these animals, suggests that they may be reservoirs of potential pathogenic antimicrobial resistant bacteria that would present a health risk to humans, but also the need to deepen our knowledge of their origin and their diffusion mechanisms in different ecological niches. The permanent absence of electricity at the sampling site, the limited time of the field missions, the tertiary nature of the forest, the wild nature of the animals that were the subject of this study, etc., are all limitations that we encountered in the field. However, the results obtained pave the way for other lines of research such as the sequencing of these MDR *E. coli* (DECs) to generate a phylogenetic tree, and then the study of the diet of these animals which could be a factor limiting the effects of these MDR *E. coli* (DECs). In addition, the study of the variation in the prevalence of these MDR *E. coli* (DECs) according to the seasons, etc., would be of great use.

## Figures and Tables

**Figure 1 pathogens-11-01082-f001:**
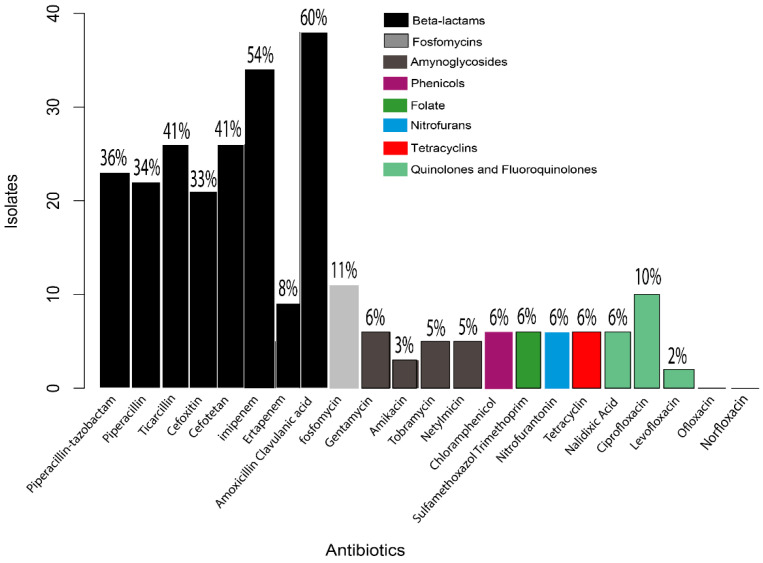
Prevalence of antimicrobial-resistant *E. coli* isolates.

**Figure 2 pathogens-11-01082-f002:**
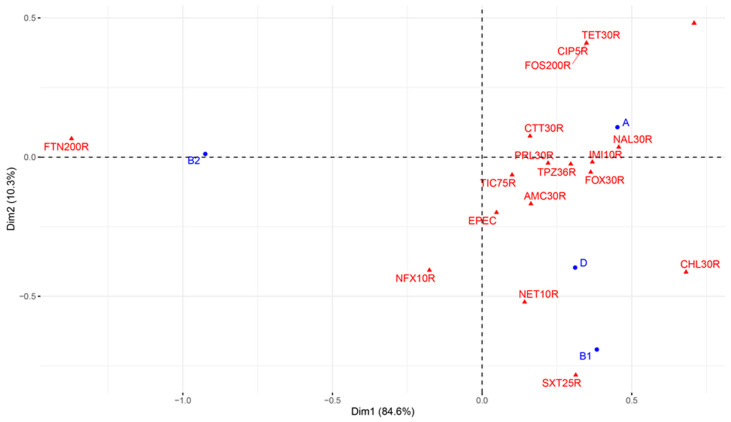
Relationship between the occurrence of EPEC virulence factors, antimicrobial resistance and phylogroups of *E. coli* isolates. This graph shows the correlation between the presence of the EPEC pathotype in gorilla *E. coli* isolates and their membership of phylogenetic groups that traditionally carry pathogenic isolates in humans, such as groups B2 and D, and the presence of antimicrobial resistance phenotypes used in human therapy. In addition, phylogenetic groups that normally carry non-pathogenic isolates, such as groups A and B1 in humans, are found to carry certain pathogenicity genes such as MDR EPEC, which is very often implicated in diarrhoeal episodes in primates, including humans and gorillas. This graph shows that there are mechanisms for transmission of MDR DECs isolates from humans to gorillas, mostly linked to the presence of phylogenetic group A.

**Table 1 pathogens-11-01082-t001:** Phylogenetic groups of *E. coli* isolates resulting from the application of the Clermont method [52,53].

Subject Studied	Sample	*chuA*	*yjaA*	*TSPE4.C2*	Phylogroup Assignation	Frequency (%)
Gorilla	Faecal	-	-	-	A (*n* = 44)	69
Gorilla	Faecal	-	+	-	A (*n* = 0)	0
Gorilla	Faecal	-	-	+	B1 (*n* = 6)	10
Gorilla	Faecal	+	+	-	B2 (*n* = 0)	0
Gorilla	Faecal	+	+	+	B2 (*n* = 13)	20
Gorilla	Faecal	+	-	-	D (*n* = 1)	1
Gorilla	Faecal	+	-	+	D (*n* = 0)	0

Legend: The phylogenetic types A (*ChuA-*, *YjaA-*, *TSPE4.C2-*), B1 (*ChuA-*, *YjaA-*, *TSPE4.C2+*), B2 (*ChuA+*, *YjaA+*, *TSPE4.C2+*) and D (*ChuA+*, *YjaA-*, *TSPE4.C2-*) have been observed in this study. In contrast, the phylogenetic types A (*ChuA-*, *YjaA+*, *TSPE4.C2-*), B2 (*ChuA+*, *YjaA+*, *TSPE4.C2-*) and D (*ChuA+*, *YjaA-*, *TSPE4.C2+*) were not observed in this study.

**Table 2 pathogens-11-01082-t002:** Pathotype groups of *E. coli* isolates.

Samples	Pathotype Group	Frequency (%)
Faecal	EPEC (*n* = 17)	85
Faecal	EPEC/EHEC (*n* = 3)	15

Legend: EPEC (enteropathogenic *E. coli*), EHEC (enterohemorrhagic *E. coli*).

**Table 3 pathogens-11-01082-t003:** Phenotypes of antibiotics, phylogroups, pathotypes and MARI [108] of *E. coli* isolated from western lowland gorilla faeces of MDNP.

Isolate	Animal	ATB	Class						Resistance Phenotype Profile								Phylogroup	Pathotype	MARI
E19	Gorilla	5	1	TIC	PIP	TZP	FOX	AMC									D	EPEC	0.23
E24	Gorilla	5	1	TIC	PIP	TZP	AMC	CTT									B2		0.23
E33	Gorilla	5	2	PIP	TZP	AMC	NAL	CTT									A		0.23
E32	Gorilla	6	3	PIP	TZP	AMC	TET	NAL	CTT								A		0.28
E40	Gorilla	6	3	PIP	TZP	IPM	AMC	FOF	TOB								A	EPEC	0.28
E41	Gorilla	6	3	PIP	TZP	IPM	TET	NAL	CTT								A		0.28
E1	Gorilla	7	2	TIC	PIP	FOX	IPM	ETP	FOF	CTT							A		0.33
E8	Gorilla	7	3	TIC	TPZ	PIP	IPM	ETP	FOF	CIP							A		0.33
E28	Gorilla	7	1	TIC	PIP	TZP	FOX	IPM	AMC	CTT							B2		0.33
E45	Gorilla	8	2	TIC	PIP	TZP	FOX	IPM	AMC	NAL	CTT						B1		0.38
E57	Gorilla	8	2	TIC	FOX	IPM	AMC	GEN	AMK	TOB	NET						A		0.38
E59	Gorilla	8	2	TIC	FOX	IPM	AMC	GEN	AMK	TOB	NET						A	EPEC	0.38
E64	Gorilla	8	6	FOX	IPM	TET	CHL	NAL	SXT	NIT	CTT						A	EPEC	0.38
E4	Gorilla	9	5	TZP	IPM	ETP	FOF	GEN	TOB	CIP	NIT	LEV					A		0.42
E7	Gorilla	9	3	TZP	IPM	ETP	AMC	FOF	GEN	AMK	NIT	LEV					A		0.42
E37	Gorilla	9	3	TIC	PIP	TZP	FOX	IPM	AMC	NAL	SXT	CTT					B1		0.42
E50	Gorilla	9	3	TIC	PIP	TZP	FOX	IPM	AMC	NET	SXT	CTT					B1		0.42
E49	Gorilla	11	6	TIC	TZP	FOX	IPM	AMC	TET	CHL	NAL	SXT	NIT	CTT			A		0.52
E63	Gorilla	12	7	TIC	PIP	TZP	FOX	IPM	AMC	NET	TET	CHL	NAL	CIP	NIT		B2	EPEC	0.57
E10	Gorilla	13	6	TIC	TZP	PIP	FOX	IPM	ETP	AMC	FOF	TOB	CHL	NAL	CIP	CTT	A		0.61

ATB = antibiotic, MARI = Multiple Antimicrobial Resistance Index, AMC = amoxicillin/clavulanic acid, TIC = ticarcillin, PIP = piperacillin, TZP = piperacillin/tazobactam, FOX = cefoxitin, ETP = ertapenem, IPM = imipenem, CTT = cefotetan, TOB = tobramycin, AMK = amikacin, NET = netilmicin, GEN = gentamycin, NAL = nalidixic acid, CIP = ciprofloxacin, NOR = norfloxacin, OFX = ofloxacin, LVX = levofloxacin, FOF = fosfomycin, TET = tetracycline, SXT = sulfamethoxazole + trimethoprim, CHL = chloramphenicol, NIT = nitrofurantoin, EPEC (enteropathogenic *E. coli*), EHEC (enterohemorrhagic *E. coli*). This table includes only resistant *E. coli* isolates with a resistance index (MARI) between 0.22–0.59 out of 64 isolates tested.

## Data Availability

Not applicable.

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
