# Peer review of "Phylogenetic Groups, Pathotypes and Antimicrobial Resistance of Escherichia coli Isolated from Western Lowland Gorilla Faeces (Gorilla gorilla gorilla) of Moukalaba-Doudou National Park (MDNP)"

_pathogens, 2022, doi:10.3390/pathogens11101082_

Round 1
Reviewer 1 Report
The manuscript by Oyaba Yinda et al. contribute with important data to the knowledge of complex puzzle of phylogenetic groups, pathotypes and antibiotic resistance of Escherichia coli isolated from gorilla feces. I support its possible publication after appropriate modifications as outlined below.
Line 3 title : Escherichia coli – italics. Please carefully revise this issue throughout the manuscript.
line 28: I strongly suggest to the authors to use antimicrobial resistant instead of „antibiotic-resistant” throughout the manuscript, it is a more appropriate term
Line 30: „Escherichia coli” instead of „E. coli”
Line 36: 67% - when you present overall prevalence values please insert in brackets the 95% confidence interval, throughout the manuscript
Line 40: „, twenty per cent” – is not in agreement with the journal requirements
Line 91: „Gorilla gorilla gorilla” – italics. Please carefully revise the writing italics of the scientific name of species throughout the manuscript, including the reference list
Line 93: „E. coli” instead of “Escherichia coli” and “spp” – not italic, please carefully revise this concern throughout the manuscript
Line 119: “… D (16%; n = 1/64), respectively.”
Lines 127-128: “high rate” and “moderate” – I wonder how the authors made these classifications?
Line 150: I suggest the inclusion of Table 1 in a supplementary file
Line 126: “Antibiotic Susceptibility testing” – was any strain multidrug or pan-drug resistant? Please specify and further discuss this concern
Table 4 “Résistance” and in legend there are some “French” characters
Line 343-343: please provide in a supplementary file the Confidence of the identification of each E. coli isolates, performed with the Vitek2 system
Line 348: The used antimicrobials selection strategy is unclear. How they were selected?
Line 345: I wonder, why chose the authors to monitor the antimicrobial susceptibility profile of E. coli isolates with Kirby–Bauer disk-diffusion method, when they have the possibility to do this with the modern Vitek2 system with nearly similar cost?
Lines 349-355: Please use standard abbreviations for antimicrobials retrieved on the following website: https://journals.asm.org/journal/aac/abbreviations
Line 355: please specify how were the results interpret. Resistant, intermediate resistant or susceptible?
Line 360: “chua, yjiaa, tspe4c2” – italics Please indicate the used positive and negative controls within the PCR reactions to validate the obtained results
Line 372: “E. coli” – italics
Line 371: The first sentence from the conclusion section is the study aim rather a conclusion
Within the conclusion section the authors must highlight the study limitations and future perspectives.
Reviewer 2 Report
The main advantage of the reviewed work is that the tests were performed on faecal samples from wild animals living in the national park. Such material is much more difficult to obtain than samples from breeding or companion animals.
However, the manuscript requires some improvement.
All Figures, Schemes and Tables should be inserted into the main text close to their first citation and must be numbered following their number of appearance. Chapter 3.5 should be deleted and tables and figures moved to the appropriate places.
Table 1 should be included in the M&M chapter, not the Results chapter.
L. 133 - There is incorrect numbering of Figures and Tables
L133 - information on the number / percentage of multi-drug resistant strains is missing; please provide the definition of multi-drug resistant strains on the basis of which the number / percentage of MDR strains was assessed
Fig. 2 - I suggest in the bar graph to include the percentage of strains resistant to a given antibiotic instead of the number of strains; the percentages give a better overview of the results obtained
Table 4 - What does the word "class" mean and the numbers in this column?
ATB - do you mean "Number of antibiotics"?
Does "TET" and "TE" mean the same?
I find the entire chapter on Results too poorly covered; there is no reference to Table 4; Table 4 lists 20 different drug resistance profiles, while the E. coli strains were 64; this should be clarified.
My reservations are raised by the methodology for determining the drug susceptibility of E. coli isolates. First, Authors should state which CLSI document they relied on - citation number 46 should be replaced with the appropriate CLSI document.
Second, there is an inconsistency between the concentrations of some antimicrobials and the current CLSI guidelines. This incompatibility applies to: ticarcillin 75ug, piperacillin 30ug, piperacyclin / tazobactam 36ug, netilmycin 10ug.
To the best of my knowledge, the current CLSI guidelines do not take into account ticarcillin 75 µg, and for other substances, the following concentrations are:
- ticarcillin / clavulanate 75/10 µg
- piperacillin 100 µg
- piperacyclin / tazobactam 100/10 ug,
- netilmycin 30 ug
I believe that the susceptibility test should be repeated using the antibiotic / chemotherapeutic concentrations indicated in the current CLSI guidelines.
P14, L349-354 - insert spaces between concentration and µg
The abstract contains imprecise information: L34 "Sixty-four E. coli isolates were obtained ..", and there is similar information in L36: Sixty-four (67%) - 64 or 67? Does this sentence mean that 64% or 67% of the collected stool samples were positive for E. coli?
L38 - What does the sentence "Twenty per cent (20%) were positive on phylogrups PCR ..." mean?
L40 - "Also, twenty per cent ..." change into 20%
L41-42 & P8 L11 - add italics
P10 L114 al. [97] - add space
P11L182 - remove the bracket; everyone knows what group of antibiotics is represented by ampicillin and streptomycin.
The Discussion chapter is too long, the authors tend to write sloppy; I suggest that you write it more concisely with the relevant facts.
P13, L271 - 272 - .... with respect to chloramphenical (add% of resistant strains), ciprofloxacin (add%), etc. [some information should be specified]
P8, L22 - The sentence "E. coli is one of the enterobacteria frequently described in wildlife" does not provide any relevant information, as it is obvious that E. coli is very widespread.
The manuscript does not refer to several important literature reports on the characteristics of E. coli strains derived from wild mammals, including those living in reservoirs or national parks, e.g.
Turchi B, Dec M, Bertelloni F, Winiarczyk S, Gnat S, Bresciani F, Viviani F, Cerri D, Fratini F. Antibiotic Susceptibility and Virulence Factors in Escherichia coli from Sympatric Wildlife of the Apuan Alps Regional Park (Tuscany, Italy). Microb Drug Resist. 2019 Jun; 25 (5): 772-780.
Bamunusinghage NPD, Neelawala RG, Magedara HP, Ekanayaka NW, Kalupahana RS, Silva-Fletcher A, Kottawatta SA. Antimicrobial Resistance Patterns of Fecal Escherichia coli in Wildlife, Urban Wildlife, and Livestock in the Eastern Region of Sri Lanka, and Differences between Carnivores, Omnivores, and Herbivores. J Wildl Dis. 2022 Apr 1; ​​58 (2): 380-383
I believe that the conclusion contains some over-interpretation that gorillas are carriers of pathogenic E. coli strains. A pathogenic strain is one that causes symptoms of the disease, but in this case we should talk about strains belonging to certain E. coli patotypes. The tested strains are potentially pathogenic, not pathogenic.
Round 2
Reviewer 1 Report
Much improved!
Author Response
We would like to take this opportunity to express our gratitude for your expertise in correcting and improving the form and content of this manuscript.
Reviewer 2 Report
The correction made by the authors is satisfactory.
I believe that the value of this manuscript would be even greater if the authors detected the genes encoding the Shiga toxin. Shiga-toxigenic E. coli (STEC) are especially dangerous to humans, and their carriers are often wild mammals. I suggest that the authors consider including stx1 and stx2 gene detection in this work.
